# Phytochemical Composition and Bioactive Potential of *Melissa officinalis* L., *Salvia officinalis* L. and *Mentha spicata* L. Extracts

**DOI:** 10.3390/foods12050947

**Published:** 2023-02-23

**Authors:** Beatriz Nunes Silva, Vasco Cadavez, Cristina Caleja, Eliana Pereira, Ricardo C. Calhelha, Mikel Añibarro-Ortega, Tiane Finimundy, Marina Kostić, Marina Soković, José António Teixeira, Lillian Barros, Ursula Gonzales-Barron

**Affiliations:** 1Centro de Investigação de Montanha (CIMO), Instituto Politécnico de Bragança, Campus de Santa Apolónia, 5300-253 Bragança, Portugal; 2Laboratório Associado para a Sustentabilidade e Tecnologia em Regiões de Montanha (SusTEC), Instituto Politécnico de Bragança, Campus de Santa Apolónia, 5300-253 Bragança, Portugal; 3CEB—Centre of Biological Engineering, University of Minho, Campus de Gualtar, 4710-057 Braga, Portugal; 4LABBELS—Associate Laboratory, 4710-057 Braga, Portugal; 5Institute for Biological Research “Siniša Stanković”—National Institute of Republic of Serbia, University of Belgrade, Bulevar Despota Stefana 142, 11000 Belgrade, Serbia

**Keywords:** Lamiaceae, polyphenols, antimicrobials, antifungals, antioxidants, anti-inflammatory effect, antiproliferative effect, alternative preservatives, biopreservation

## Abstract

Plants are rich in bioactive phytochemicals that often display medicinal properties. These can play an important role in the production of health-promoting food additives and the replacement of artificial ones. In this sense, this study aimed to characterise the polyphenolic profile and bioactive properties of the decoctions, infusions and hydroethanolic extracts of three plants: lemon balm (*Melissa officinalis* L.), sage (*Salvia officinalis* L.) and spearmint (*Mentha spicata* L.). Total phenolic content ranged from 38.79 mg/g extract to 84.51 mg/g extract, depending on the extract. The main phenolic compound detected in all cases was rosmarinic acid. The results highlighted that some of these extracts may have the ability to prevent food spoilage (due to antibacterial and antifungal effects) and promote health benefits (due to anti-inflammatory and antioxidant capacities) while not displaying toxicity against healthy cells. Furthermore, although no anti-inflammatory capacity was observed from sage extracts, these stood out for often displaying the best outcomes in terms of other bioactivities. Overall, the results of our research provide insight into the potential of plant extracts as a source of active phytochemicals and as natural food additives. They also support the current trends in the food industry of replacing synthetic additives and developing foods with added beneficial health effects beyond basic nutrition.

## 1. Introduction

Over recent years, with the increasingly negative perception of consumers towards artificial food additives [1] and the higher demand for nutritious foods with additional health benefits, two major trends in the food industry have been to replace synthetic additives, which may be harmful to human health [2], and to develop nutraceuticals/functional foods [3].

In line with these trends, modern science has shown that plant matrices are sources of valuable molecules (for example, phenolic compounds) with promising biological value (e.g., antioxidant, anti-inflammatory, antibacterial and antifungal), thus encouraging their use for the development of functional foods and nutraceuticals, and as possible substitutes for artificial additives in foods or their packaging [4,5,6]. However, it is necessary to guarantee that the herbal extracts are safe for human consumption, and, among other considerations, it is crucial that they are obtained: (i) using nontoxic solvents authorised for the industrial production of foodstuffs and food ingredients, which do not leave residues or derivatives in the product after removal (or leave them in technically unavoidable quantities that pose negligible risk to human health) [7,8]; and (ii) from herbs with documented traditional use, commonly used in cooking as aroma and/or flavour enhancers [6,8,9].

To this, lemon balm (*Melissa officinalis* L., Lamiaceae), sage (*Salvia officinalis* L., Lamiaceae) and spearmint (*Mentha spicata* L., Lamiaceae) are among the various plants widely used in traditional Mediterranean cuisine and medicine, and for which several researchers have reported health-promoting capacities and potential as natural food additives [10,11,12]. Lemon balm has many beneficial capacities, such as spasmolytic, sedative, antitumoral, antimicrobial and antioxidant effects [13]. Furthermore, this plant has shown therapeutic effects for the treatment of the cognitive disturbance of Alzheimer’s disease, and has been traditionally used to reduce anxiety, sleep disturbance, depression and gastrointestinal disorders [13,14]. In relation to sage, this herb has been used as a gargle for throat inflammations, to reduce perspiration, improve regularity of menstrual cycle, decrease hot flashes in menopause, battle gastrointestinal problems, prevent neurodegenerative diseases and improve mental capacity [11,15]. Furthermore, sage has shown anti-inflammatory, antimicrobial, hypoglycemic, antidiabetic, antioxidant and antitumor activities [15]. As for spearmint, it is frequently used in folk medicine against gastrointestinal and respiratory complications, haemorrhoids, stomach ache, memory dysfunction, and can be used as a carminative, antispasmodic, diuretic, antibacterial, antifungal and antioxidant agent [12,16,17].

Considering the recognised beneficial effects for human health of lemon balm, sage and spearmint, the goal of this research was to chemically characterise and appraise the bioactivities of extracts from such plants, produced through different environmentally friendly extraction methods (decoction, infusion and maceration), using water and 80% ethanol (*v*/*v*) as solvents. More specifically, the extracts’ cytotoxicity, antibacterial, antifungal, anti-inflammatory and antioxidant capacities were evaluated to assess their safety and preservative effects.

## 2. Materials and Methods

### 2.1. Plant Material and Extraction Procedures

Sage, lemon balm and spearmint dry aerial parts were supplied by Pragmático Aroma, Lda. (“Mais Ervas”, Trás-os-Montes, Portugal), mechanically milled and submitted to the following extraction methods: infusion, decoction and dynamic maceration.

For the infusions, 2 g of plant material was mixed with 200 mL of boiling distilled water and left to rest for 5 min without additional heating. For the decoctions, 2 g of plant material was mixed with 200 mL of distilled water, heated to boiling and boiled for 5 min. Infusions and decoctions were then filtrated (7–10 μm), frozen and lyophilised (FreeZone 4.5, Labconco, Kansas City, MO, USA). To obtain hydroethanolic extracts, dynamic macerations were conducted by incorporating 1 g of plant material in 30 mL of ethanol at 80% (*v*/*v*) and stirring at room temperature for 1 h. The supernatants were filtrated (7–10 μm), another 30 mL of ethanol 80% (*v*/*v*) was added to the extraction residues, and the maceration was repeated for 1 h. Finally, the ethanolic portion was evaporated (Büchi R-210, Flawil, Switzerland) and the resulting extracts were frozen and lyophilised. The extractions were carried out in triplicate (n = 3).

### 2.2. Identification and Quantification of Individual Phenolic Compounds

Individual phenolic compounds were investigated using a previously validated method, as described by Restivo et al. [18]. First, the samples were dissolved in ethanol 20% *(v*/*v*) up to a final concentration of 10 mg/mL and filtered through disposable 0.22 µm filters. The phenolic profiles were then determined by a liquid chromatography system (Dionex Ultimate 3000 UPLC, Thermo Scientific, San Jose, CA, USA) equipped with a quaternary pump, an automatic injector at 5 °C, a degasser and a column compartment with an automated thermostat. Compound detection was carried out with a diode-array detector (at wavelengths of 280 nm, 330 nm and 370 nm), coupled to a mass spectrometry (MS) detector. Separation was performed on a reverse phase Waters Spherisorb S3 ODS-2 C18 column (4.6 mm × 150 mm, 3 µm) at 35 °C. The flow rate was 0.5 mL/min. The mobile phase used was water/formic acid 0.1% (A) and acetonitrile (B). The elution gradient for solvent B was as follows: 10–15% eluent B up to 5 min, 15–20% B up to 5 min, 20–25% B 10 min, 25–35% B 10 min, 35–50% B 10 min and column re-equilibration for 10 min. For MS detection, a Linear Ion Trap LTQ XL spectrophotometer equipped with an electrospray ionization source was used. Nitrogen (50 psi) was used as a carrier gas, and the system worked with an initial temperature of 325 °C, a spray voltage of 5 kV and a capillary voltage of −20 V. The tube lens offset voltage remained at −66 V. Spectra were recorded in negative ion mode 100–1500 *m*/*z*. 

The phenolic compounds were identified through their chromatographic characteristics by comparison to the obtained standard compounds (4-hydroxybenzoic acid, apigenin-6-*C*-glucoside, apigenin-7-*O*-glucoside, caffeic acid, chlorogenic acid, naringenin and rosmarinic acid) and with the literature [19,20,21]. For quantitative analysis, calibration curves prepared with appropriate standards (between 100 and 2.5 mg/L) were used. Limits of detection and quantification were also calculated, and, in all cases, the coefficient of linear correlation was R^2^ > 0.99 (Appendix A). All analyses were made in triplicate (n = 3). The results were expressed in mg per g of dry extract (mg/g).

### 2.3. Biological Evaluation

#### 2.3.1. Antibacterial and Antifungal Activity

For the antibacterial and antifungal activity screening, six bacterial strains were used: *Escherichia coli* (ATCC 25922), *Salmonella enterica* serovar Typhimurium (ATCC 13311), *Enterobacter cloacae* (clinical isolate), *Staphylococcus aureus* (ATCC 11632), *Bacillus cereus* (food isolate) and *Listeria monocytogenes* (NCTC 7973), and six micromycetes: *Aspergillus fumigatus* (human isolate), *Aspergillus niger* (ATCC 6275), *Aspergillus versicolor* (ATCC 11730), *Penicillium funiculosum* (ATCC 36839), *Penicillium verrucosum* var*. cyclopium* (food isolate) and *Trichoderma viride* (IAM 5061). 

Minimum inhibitory (MIC), minimum bactericidal (MBC) and minimum fungicidal (MFC) concentrations were determined using a broth microdilution method and 96-well microplates [22]. The streak plate culture method, conducted on tryptic soy agar (Torlak, Belgrade, Serbia) incubated at 37 °C for 24 h, was used to obtain bacterial cells in the exponential growth phase. Then, an adequate number of individual colonies were placed in tubes with sterile water to achieve bacterial suspensions with a concentration of approximately 1.0 × 10^5^ CFU/well in the microplates. For the antifungal activity essay, fungal spores were washed from the surface of malt agar plates (Neogen, Heywood, UK) with sterile 0.85% saline added with 0.1% Tween 80 (*v*/*v*) (Zorka pharma, Šabac, Belgrade). Sterile saline was then used to adjust the spore suspension to a concentration of approximately 1.0 × 10^5^ in a final volume of 100 µL per well.

For the antibacterial and antifungal essay, resuspended extracts were obtained by dissolving them in ethanol 30% (*v*/*v*) to obtain a final concentration of 10 mg/mL. The liquid media (90 μL) used in the microplate wells was tryptic soy broth (Torlak, Belgrade, Serbia) for the antibacterial essay, or malt extract broth (Neogen, Heywood, UK) in the case of the antifungal essay.

After placing the inoculum, resuspended extract and liquid media in the microplate wells as appropriate, the microdilution plates were incubated at 37 °C for 24 h for the determination of the antibacterial activity, or 28 °C for 72 h for the determination of the antifungal activity. After that, 40 μL of iodonitrotetrazolium (Sigma-Aldrich, St. Louis, MO, USA), at a concentration of 0.2 mg/mL, was added to each well, and the microplate incubated again at 37 °C for 1 h. Afterwards, the microplates were evaluated, and the lowest concentrations without visible growth were determined as the MICs. The MBCs were determined as the lowest concentration with no visible growth after serial sub-cultivation of 10 µL into microdilution plates containing 100 µL of tryptic soy broth per well and further incubated for 24 h at 37 °C.

For the antifungal essay, MICs were determined under binocular microscope using the same procedure as described above. After that, the MFC was determined by serial sub-cultivation of 2 µL of the content of the wells and further incubation at 28 °C for 72 h. The lowest concentration of this sub-culture with no visible growth was defined as the MFC.

Two commonly used artificial food preservatives, sodium benzoate (E211) and potassium metabisulfite (E224), were also tested to evaluate the sensitivity of the microorganisms to such additives. The MIC, MBC and MFC were expressed in mg/mL of the resuspended lyophilised extracts.

#### 2.3.2. Antioxidant Activity

The antioxidant activity was evaluated through two in vitro essays, using previously described methodologies [23,24]: inhibition of lipid peroxidation by decrease in the formation of thiobarbituric acid reactive substances (TBARS), and the oxidative haemolysis inhibition essay (OxHLIA). The extracts were initially diluted in distilled water (for TBARS) or phosphate-buffered saline (PBS, pH 7.4) (for OxHLIA) to different concentrations. Trolox was used as a positive control in both essays.

For TBARS essay: the extracts were examined for their power to inhibit the ferrous sulphate-induced lipid peroxidation, using porcine brain cell homogenates, through monitorisation of the colour strength (at 532 nm) provided by malondialdehyde-thiobarbituric acid complexes. The results were expressed as the extract concentration (μg/mL) required to inhibit 50% of the TBARS formation (half-maximal inhibitory concentration, IC_50_).

For OxHLIA essay: 200 µL of an erythrocyte solution at 2.8% prepared in PBS was added to 400 µL of either: extract solution (13–800 μg/mL in PBS), PBS solution (negative control), distilled water (baseline), or Trolox (7.81–250 µg/mL). After incubation for 10 min at 37 °C with agitation, 200 μL of 2,2′-azobis(2-methylpropionamidine) dihydrochloride (AAPH, 160 mM in PBS) was added, and the optical density (at 690 nm) was measured in a microplate reader (Bio-Tek Instruments, ELx800, Winooski, VT, USA) every 10 min until complete haemolysis. The percentage of the erythrocyte population that remained undamaged (*P*) was calculated using Equation (1), where *S_t_* and *S*_0_ are the optical density of the sample at *t* and 0 min, respectively, and *CH*_0_ is the optical density of the complete haemolysis at 0 min.
(1)$$P\%=100\times \left(\frac{S_{t}-CH_{0}}{S_{0}-CH_{0}}\right)$$

The delayed time of haemolysis (Δ*t*) was calculated using Equation (2), where the 50% haemolytic time (min) graphically obtained from the haemolysis curve of each sample concentration is represented by H*t*_50_:
Δ*t* (min) = H*t*_50_ (sample) − H*t*_50_ (control) (2)

Lastly, the Δ*t* values were correlated to the various sample concentrations. From that correlation, the concentrations able to promote Δ*t* haemolysis delays of 60 min and 120 min were calculated. The results were expressed as IC_50_ values (μg/mL) at Δ*t* = 60 min and Δ*t* = 120 min, i.e., the sample concentration required to protect 50% of the erythrocyte population from the haemolytic action of AAPH for 60 min and 120 min, respectively.

#### 2.3.3. Anti-Inflammatory Activity

The anti-inflammatory activity was evaluated using a previously described essay, with modifications [25]. First, cells from the mouse macrophage-like cell line RAW264.7 were seeded in plates of 96-wells, and their attachment was allowed overnight. Subsequently, cells were subjected to different extract concentrations (6.25–400 μg/mL) for 1 h, and then stimulated with lipopolysaccharides (1 μg/mL) for 18 h. This procedure enabled observation of the occurrence of induced changes in nitric oxide basal levels, using a Griess Reagent System kit (Promega, Madison, WI, USA). The nitrite level produced was determined in a microplate reader (Bio-Tek Instruments, ELx800, Winooski, VT, USA) by assessing the optical density at 540 nm and comparing it with the standard calibration curve. The positive control used was dexamethasone (50 μM). The results are stated as the sample concentration (μg/mL) necessary to inhibit 50% of the nitric oxide production (IC_50_).

#### 2.3.4. Cytotoxic Activity

The lyophilised extracts were dissolved in water and successively diluted to obtain the stock solutions. The cytotoxic activity was then assessed against six human tumour cell lines, namely AGS (gastric adenocarcinoma), CaCo-2 (colorectal adenocarcinoma), HeLa (cervical carcinoma), MCF-7 (breast adenocarcinoma), NCI-H460 (large cell lung carcinoma) and non-tumour hFOB (human foetal osteoblasts), using the previously described sulforhodamine B essay [25]. For this, each of the cell lines (190 µL, 10^4^ cells/mL) was incubated with the plant extracts at various concentrations (6.25–400 µg/mL). Ellipticine was used as a positive control. The results were expressed as the extract concentration required to inhibit 50% of the net cell growth (half-maximal cell growth inhibitory concentration, GI_50_). 

### 2.4. Statistical Analysis

Data were presented as mean ± standard deviation (SD) values. One-way analysis of variance (ANOVA, α = 0.05) was used to assess statistical differences between the means. Clustered heatmaps were generated using the pheatmap function from the pheatmap package [26]. Statistical analysis was conducted in R software (version 4.1.0, R Foundation for Statistical Computing, Vienna, Austria).

## 3. Results and Discussion

### 3.1. Phenolic Profile

The peak characteristics (retention time, wavelength of maximum absorption and mass spectral data), tentative identification and quantification of the phenolic compounds detected in the extracts produced are reported in Appendix A of the Appendix A (sage, lemon balm and spearmint, respectively). Heatmaps for a fast visualisation of the phenolic compounds identified and their concentrations were produced and are shown in Figure 1, Figure 2 and Figure 3 (sage, lemon balm and spearmint, respectively).

The dendrograms of each clustered heatmap arrange the information on phenolic composition in terms of similarities, where the lower the height at which any two objects are joined, the greater the similarity. In this sense, one dendrogram (left) offers insight regarding compounds detected in similar concentrations across extracts obtained through different methodologies (infusion, decoction and hydroethanolic maceration), whereas the other dendrogram (upper) informs about similar total phenolic compound content across the extracts produced, for each plant.

Twenty-four phenolic compounds were identified in all sage extracts. From Figure 1 and Appendix A, sage decoction and infusion contained higher and similar total phenolic compounds content (84.07 and 77.67 mg/g extract, respectively), compared to the hydroethanolic extract (63.17 mg/g extract). In the case of lemon balm, a maximum of fourteen compounds were identified, depending on the extract type. Figure 2 and Appendix A indicate that its infusion and hydroethanolic extract showed comparable total phenolic compounds content (61.00 and 58.35 mg/g extract, respectively); however, lower than that of the decoction (84.51 mg/g extract). As for the spearmint extracts, a maximum of fourteen compounds were identified. Figure 3 and Appendix A reveal that spearmint infusion and hydroethanolic extract had closer total phenolic compounds concentration (38.79 and 57.92 mg/g extract, respectively) than spearmint decoction (77.20 mg/g extract). Considering these results, decoctions revealed the highest amount of total phenolic compounds when compared to infusions and hydroethanolic extracts, regardless of the plant (Appendix A). Overall, sage and lemon balm decoctions stood out for their higher total phenolic content (84.07 and 84.51 mg/mL, respectively). Oppositely, spearmint infusion yielded the lowest total phenolic content among the nine extracts (38.79 mg/g extract, Appendix A).

In all cases, the plant extracts revealed a higher content of total phenolic acids compared with total flavonoids (Appendix A). This was particularly noticeable in lemon balm extracts, which presented total flavonoid concentrations lower than 0.7 mg/g extract, in comparison with the total phenolic acids content, which ranged between 57.74 and 83.90 mg/g extract. In terms of qualitative profile, sage showed the highest variety of phenolic acids, with a total of fifteen different acids regardless of the type of extract. In comparison, in lemon balm, twelve or thirteen distinct phenolic acids were identified, depending on the extract type, whereas nine or ten acids were identified in spearmint extracts.

Some phenolic acids were found across all the evaluated extracts, namely, rosmarinic and salvianolic acids, as well as lithospermic acid A. Among these, the major compound in all the sage extracts was *cis*-rosmarinic acid (22.72 to 28.40 mg/g extract; Figure 1 and Appendix A), followed by a derivative of luteolin, luteolin-7-*O*-glucuronide (12.06 to 18.07 mg/g extract; Figure 1 and Appendix A). In lemon balm, rosmarinic acid was found in the greatest amount, irrespective of the type of extract, with concentrations between 34.40 and 41.71 mg/g extract (Figure 2 and Appendix A). Similarly, the major phenolic compound in spearmint extracts was rosmarinic acid (19.61 to 32.08 mg/g extract; Figure 3, Appendix A). Rosmarinic acid is known to possess extraordinary therapeutic potential, which includes antiviral, antibacterial, anticarcinogenic, antioxidant, anti-aging, antidiabetic, cardioprotective, hepatoprotective, nephroprotective, antidepressant, antiallergic and anti-inflammatory activities [27].

Sage extracts presented the highest number of different flavonoids (nine in total). These were derivates of apigenin and luteolin, with the most abundant compound being luteolin*-*7-O-glucuronide. Flavonoids were also detected in lemon balm and spearmint extracts, although in lesser variety (one and four in total, respectively), and these were also luteolin derivatives. 

Given these results, sage, lemon balm and spearmint extracts appear to be valuable sources of valuable bioactive compounds, particularly of phenolic acids. Previous studies also investigated the phenolic profile of the plant materials used in this work. In sage and lemon balm hydroethanolic extracts, Spréa et al. [6] identified twenty-one and twelve phenolic compounds, respectively, several of which were also detected in the present study. Maliki et al. [28] studied the polyphenolic profile of a sage aqueous extract, identifying eighteen compounds, most of which belonged to hydroxycinnamic acid, rosmarinic acid and luteolin derivatives. Both the studies of Spréa et al. and Maliki et al. [6,28] found rosmarinic acid (51.00 mg/g and 2.192 mg/g, respectively) and luteolin-7-*O*-glucuronide (27.00 and 1.877 mg/g, respectively) to be the compounds of the highest concentrations in sage extracts, thus supporting the findings of our study. Also, in agreement with our results, Cirlini et al. [12] identified rosmarinic acid and its derivatives as the most prevalent polyphenolic compounds in an aqueous spearmint extract (230.5 mg/g), followed by salvianolic acids (14.70 mg/g) and caffeoylquinic acids (3.06 mg/g). Silva et al. [29] identified rosmarinic acid as the main compound in aqueous (204 mg/L) and hydroethanolic (333 mg/L) spearmint extracts; however, in lemon balm hydroethanolic extract, naringin was the principal compound (116 mg/L), and in sage aqueous and hydroethanolic extracts, hesperidin was present in the greatest amount (279 and 805 mg/L, respectively). This and other studies may have reported different phytochemical compositions [30,31], which however does not conflict with our results, since variations can be caused by different environmental factors during plant development, including soil type, change in season, salinity, light, altitude and humidity, as well as plant growth stage and extraction procedure [12,32]. Since the health-promoting properties of plants have been largely attributed to their phenolic compounds (among other secondary metabolites) [33,34], it is intuitive that differences in phenolic profile among extracts produced from the same plant matrix will also originate variations in their bioactivities (antimicrobial, antioxidant and anti-inflammatory, for example).

### 3.2. Antibacterial and Antifungal Activity

The results of the antibacterial and fungicidal activity are shown in Table 1 and Table 2, respectively.

Overall, the extracts revealed antimicrobial activity against all foodborne pathogens tested, namely *S. aureus*, *B. cereus*, *L. monocytogenes*, *E. coli*, *S.* Typhimurium and *E. cloacae* (MIC ≤ 2 mg/mL; MBC ≤ 4 mg/mL). Sage infusion presented the lowest MIC and MBC values of all extracts (i.e., the greatest antimicrobial potential), particularly against *S. aureus* and *B. cereus* (MIC = 0.25 and MBC = 0.5 mg/mL in both cases). On the other hand, lemon balm decoction displayed the highest MIC and MBC values, specifically against *L. monocytogenes* (MIC = 2 and MBC = 4 mg/mL). With a few exceptions, hydroethanolic extracts showed uniform activity (MIC = 0.5 and 1 mg/mL) for all tested bacteria.

In terms of antifungal capacity, all the infusions and hydroethanolic extracts were effective in inhibiting the six fungi tested, *A. fumigatus*, *A. niger*, *A. versicolor*, *P. funiculosum*, *P. verrucosum* and *T. viride* (MIC ≤ 1 mg/mL; MFC ≤ 2 mg/mL). Infusions demonstrated inhibition activity against the tested fungi with MIC values between 0.125 and 0.5 mg/mL, except for spearmint infusion against *P. verrucosum* var. *cyclopium* (MIC = 1 mg/mL*)*. Hydroethanolic extracts stood out for inhibiting *T. viride* at a low concentration (MIC = 0.125 mg/mL for spearmint and sage extracts; MIC = 0.25 mg/mL for lemon balm extract), which demonstrates the susceptibility of this microorganism to such extracts. The three decoctions were also effective against all fungi (MIC ≤ 0.5 mg/mL; MFC ≤ 1 mg/mL) except *A. niger* (MIC > 4 mg/mL for lemon balm and spearmint).

In general, the infusions, decoctions and hydroethanolic extracts showed comparable or higher antimicrobial and fungicidal activities than those of the artificial food preservatives E211 and E224. In particular, the results of E211 against *S. aureus* (MIC and MBC = 4 mg/mL) and *P. verrucosum* (MIC = 2 and MFC = 4 mg/mL), and those of E224 against *B. cereus* (MIC = 2 and MBC = 4 mg/mL) differ noticeably from the lower concentration of plant extracts needed to prevent the growth of such microorganisms. These findings point out the potential of the extracts tested in this study as good candidates for applications in food and possible alternatives for replacing synthetic preservatives, aiming to delay the proliferation of food spoilage and pathogenic bacteria and fungi.

In line with our research, some previous studies have also reported on the antimicrobial and antifungal effects of these plants. The sage infusions of Abdel-Wahab et al. [35] showed MIC values of 50 mg/mL for *E. coli*, and 75 mg/mL for *Salmonella* spp., *S. aureus* and *B. cereus*. Hydroethanolic sage extracts produced by Hemeg et al. [36] revealed MIC values of 5 mg/mL for *S. aureus*, 0.625 mg/mL for *B. cereus* and 2.5 mg/mL for *E. coli* and *S.* Enteritidis. Silva et al. [29] hydroethanolic sage extracts revealed MIC values of 2.5–5 mg/mL for *L. monocytogenes*, 0.625 mg/mL for *S. aureus*, 10 mg/mL for *S.* Typhimurium and 1.25 mg/mL for *E. coli*. In turn, Ueda et al. [37] investigated hydroethanolic sage extracts obtained through ultrasound-assisted extraction, and MIC values were 1 mg/mL for *S. aureus*, *B. cereus*, *L. monocytogenes*, *E. coli*, *S.* Typhimurium and *E. cloacae*, 0.25 mg/mL for *A. fumigatus*, *A. versicolor*, *P. funiculosum* and *P. verrucosum* and 0.5 mg/mL for *A. niger* and *T. viride*.

Silva et al. [29] also tested the hydroethanolic extracts of spearmint and lemon balm, which revealed MIC values of 2.5 mg/mL for *L. monocytogenes*, 1.25 mg/mL for *S. aureus*, 20 mg/mL for *S.* Typhimurium and 1.25 mg/mL for *E. coli* for spearmint, and 5 mg/mL for *L. monocytogenes*, 2.5 mg/mL for *S. aureus*, 20 mg/mL for *S.* Typhimurium and 2.5 mg/mL for *E. coli* for lemon balm. Caleja et al. [38] analysed the antimicrobial activity of spearmint infusions, reporting MIC values of 0.5 mg/mL for *L. monocytogenes*, *B. cereus* and *E. coli* and 0.25 mg/mL for *S.* Typhimurium. The same study also determined the MIC of lemon balm infusions, which revealed values of 1 mg/mL for all bacteria mentioned before [38]. Furthermore, Caleja et al. [38] evaluated the MIC of said infusions against *A. niger*, *A. versicolor*, *P. funiculosum* and *P. verrucosum*, and the values ranged between 0.25 and 1 mg/mL.

### 3.3. Antioxidant Activity

The results of the TBARS and OxHLIA essays, which assess the ability of the plant extracts to inhibit lipid peroxidation and oxidative haemolysis in vitro, are presented in Table 3. The results are expressed as IC_50_ values, meaning that lower values correspond to greater antioxidant potential.

In both TBARS and OxHLIA essays, the antioxidant capacity of each plant infusion was significantly different from that of the other two (*p* < 0.05). Differences were also found among the decoctions, in both essays, depending on the plant species (*p* < 0.05). The antioxidant power of the hydroethanolic extracts also displayed differences depending on the plant used (*p* < 0.05), although not all of them were significant in the case of the OxHLIA essay. Moreover, in both essays, for each plant, different extraction methods yielded distinct antioxidant activities (*p* < 0.05). The exception was the decoction and hydroethanolic extract of lemon balm, which presented similar antioxidant potential in the TBARS essay (*p* > 0.05).

Overall, according to the statistical analysis, lemon balm infusion and sage hydroethanolic extract (125 µg/mL and 132 µg/mL, respectively) showed the best capacities to inhibit the formation of malondialdehyde and other reactive substances that are the result of the ex vivo decomposition of lipid peroxidation products (in the TBARS essay).

The results of the OxHLIA essay showed that the sage decoction (8.93 µg/mL and 23.5 µg/mL, for Δ*t* = 60 min and 120 min) and the hydroethanolic extracts of spearmint (12.5 µg/mL and 27.6 µg/mL, for Δ*t* = 60 min and 120 min) and of lemon balm (13.5 µg/mL and 27.4 µg/mL, for Δ*t* = 60 min and 120 min) exhibited the greatest antioxidant protection for the erythrocyte membrane, even compared to the pure antioxidant compound used as a positive control, Trolox (21.8 µg/mL and 43.5 µg/mL, for Δ*t* = 60 min and 120 min). These results suggest the potential of such extracts to be used against free radical-induced oxidative damage of biological membranes.

Furthermore, the OxHLIA essays allows us to distinguish between short-term and long-term antioxidant protection, as the antioxidant behaviour is monitored over time and the oxidative haemolysis assessed at two Δ*t*. It was observed that all the infusions had anti-haemolytic activity for longer exposure times, as the concentration necessary to protect 50% of the red blood cells for 120 min was less than double the concentration necessary for this protection for 60 min. This also occurred in the case of spearmint and lemon balm decoctions, but not for the remaining extracts. 

Our findings agree with other researchers that have also reported on the antioxidant capacities of lemon balm, spearmint and sage. Groupwise summary statistics calculated by Silva et al. [29] showed the high antioxidant power of these three plants, determined by the free radical scavenging (DPPH), radical cation decolorization (ABTS) and ferric reducing antioxidant power (FRAP) essays: the results were between 259 and 507 µmol Trolox Equivalent/g dry plant, for the DPPH and ABTS essays, and between 722 and 1013 µmol Fe^2+^/g dry plant for the FRAP essay. Abdel-Wahab et al. [35] also evaluated a sage extract, using the DPPH method, and reported an IC_50_ of 13.34 µg/mL. Ueda et al. [37] reported an IC_50_ of 2.6 mg/g of sage extract, determined by the OxHLIA method, for the time period of 120 min. Caleja et al. [38] used two methods to assess the antioxidant power, reporting IC_50_ values of 6.6 µg/mL and 4.2 µg/mL for lemon balm and spearmint extracts, respectively (using the TBARS essay), and IC_50_ values of 24.8 µg/mL and 38.3 µg/mL for lemon balm and spearmint extracts, respectively (using the OxHLIA method for Δ*t* = 60 min).

### 3.4. Anti-Inflammatory Activity

Table 4 presents the anti-inflammatory activity essay results. These are expressed as IC_50_ values, so lower values correspond to greater anti-inflammatory potential.

The outcomes shown in Table 4 indicate that most extracts did not reveal anti-inflammatory action at the tested concentrations (IC_50_ > 400 µg/mL). Only those of spearmint showed this capability, regardless of the extraction method. Spearmint hydroethanolic extract showed the greatest anti-inflammatory capacity, considering its IC_50_ of 26.6 µg/mL. 

In agreement with our results, the spearmint infusions of Caleja et al. [38] also displayed anti-inflammatory activity against the RAW 246.7 cell line (IC_50_ = 324 µg/mL), whereas those of lemon balm did not (IC_50_ > 400 µg/mL).

Nonetheless, and despite our results, some researchers have reported anti-inflammatory effects of sage and lemon balm extracts, meaning that these plants may be capable of offering such beneficial capacity under different circumstances [39,40].

It could be expected that extracts with high rosmarinic acid concentrations and promising antioxidant activity (low IC_50_ values in Table 3), such as sage or lemon balm infusions, for example, would also show anti-inflammatory potential, as antioxidants can reduce the inflammatory process caused by the overproduction of free radicals [25]. However, from the results in Table 4, it is noticeable that extracts presenting anti-inflammatory activity were not always the ones with the highest antioxidant capacity (except for spearmint hydroethanolic extract, which presented the lowest IC_50_ = 12.5 µg/mL in the OxHLIA essay among that type of extract). In this sense, it is important, when conducting analyses, to evaluate all bioactivities, and not to infer the results of one essay from the outcomes of another, to avoid arriving at wrongful conclusions, or even discarding plant extracts with substantial potential in terms of one particular bioactivity.

### 3.5. Cytotoxic Activity

The cytotoxicity essay results are shown in Table 5. These are expressed as GI_50_, meaning that lower outcomes correspond to greater cytotoxic capacity. 

All nine extracts produced revealed inhibitory potential (GI_50_ < 400 µg/mL) against at least one tumour cell line. Overall, the extracts were more active in tumour cells AGS, CaCo-2, HeLa and MCF-7 than NCI-H460. In fact, the cytotoxic capacity of the infusions and decoctions in the NCI-H460 tumour line was non-existent (GI_50_ > 400 µg/mL); however, some hydroethanolic extracts revealed activity.

The absence of toxicity (GI_50_ > 400 µg/mL) against non-tumour human foetal osteoblast cells, hFOB, was evident in the case of infusions and two decoctions (the exception was that of sage), which is a desirable outcome as extracts to be used in food products must be safe for consumption and cannot display toxicity against healthy cells. In contrast, the majority of hydroethanolic extracts (except that of sage, curiously) showed a cytotoxic effect towards hFOB cells, suggesting that this methodology may induce toxicity to the extracts, thus compromising their applicability as food additives.

From all the extracts, those that are non-toxic against hFOB and simultaneously present inhibitory potential against AGS, CaCo-2, HeLa and MCF-7 cells are: sage hydroethanolic extract and the infusions of lemon balm, spearmint and sage. These results point out the cytotoxic potential of the infusions produced in comparison to other extraction methods. The infusion of spearmint, specifically, showed overall greater antiproliferative capacity, with GI_50_ values of 196 µg/mL for the AGS cell line, 304 µg/mL for the CaCo-2 cell line, 229 µg/mL for the HeLa cell line and 203 µg/mL for the MCF-7 cell line.

The results obtained in this study agree, to some extent, with those of other researchers. Sage hydroethanolic extracts produced by Ueda et al. [37] did not show hepatotoxicity in PLP2 cells (non-tumour) at the maximum tested concentration of 400 µg/mL. Lemon balm and spearmint infusions of Caleja et al. [38] did not show toxicity for non-tumour cells PLP2 (GI_50_ > 400 µg/mL) and inhibited the growth of the HeLa cell line (GI_50_ = 241 µg/mL and GI_50_ = 251 µg/mL, respectively), in agreement with our results. Their spearmint infusion also inhibited MCF-7 growth (GI_50_ = 283 µg/mL), as found in our study. However, in contrast to our findings, lemon balm and spearmint infusions were able to inhibit NCI-H460 (GI_50_ = 290 µg/mL and GI_50_ = 322 µg/mL, respectively), and lemon balm infusion was incapable of affecting MCF-7 viability (GI_50_ > 400 µg/mL) [38].

Overall, these results indicate that extracts originating from any of the plants examined are potentially valuable for their cytotoxic impact on various tumour cell lines. However, it is crucial to further evaluate potential undesired effects against healthy cell lines, as even reduced concentrations may result in dangerous consequences for human health. 

## 4. Conclusions

This work revealed the biological capacities of sage, spearmint and lemon balm extracts. Although only spearmint extracts showed anti-inflammatory potential, all infusions, decoctions and hydroethanolic extracts presented encouraging results in terms of antibacterial, antifungal and antioxidant capacities. Infusions revealed the most promising results, compared to decoctions and hydroethanolic extracts, as they yielded the best outcomes in each of the essays conducted (antimicrobial, antioxidant, anti-inflammatory and antiproliferative tests), while displaying an absence of toxicity against non-tumour cells, and even though infusions did not contain the highest total phenolic contents. Extracts from sage stood out from the remainder as they were often among those presenting the best capacities, both in terms of inhibiting the oxidation and growth of pathogenic bacteria and fungi, as well as impairing the viability of tumour cells. Nonetheless, no anti-inflammatory action was detected.

Overall, the results of this study emphasise the potential value of sage, spearmint and lemon balm extracts as natural food ingredients to prevent spoilage, provide beneficial health effects and potentially replace artificial additives, hence aligning with current trends in the food industry. However, further in vitro and in vivo studies must be conducted to verify the functionality of these extracts: for example, evaluating their pharmacokinetic parameters (bioavailability and bioaccessibility). It is also expected that the food matrix has some impact on the bioactivities of plant extracts, causing differences between the results observed in vitro and in vivo, which may limit the bio-functionalities of such extracts in food products. Another obstacle that must be investigated and that herbal extracts may face is related to their effect on the sensory characteristics of foods, since the concentrations necessary to provide the desired biological capacities can be very high and, therefore, negatively affect the aroma and taste of the products. In this sense, further research must be conducted to complement in vitro studies and address these and other limitations. 

## Figures and Tables

**Figure 1 foods-12-00947-f001:**
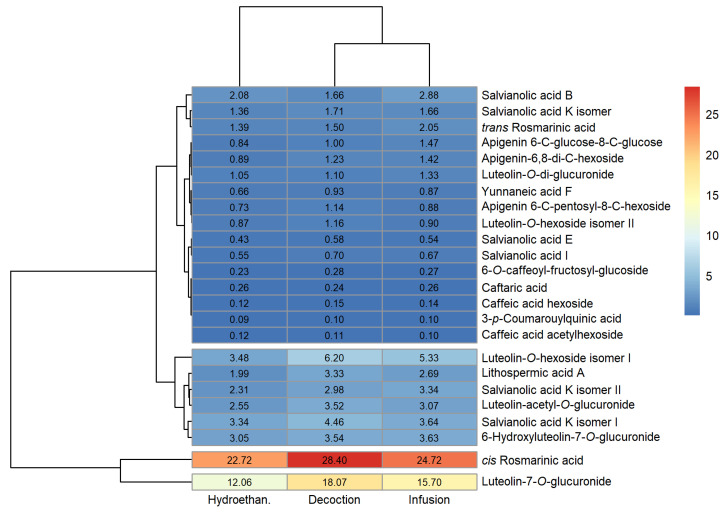
Clustered heatmap visualisation of phenolic compounds detected in sage infusion, decoction and hydroethanolic extract (units: mg/g).

**Figure 2 foods-12-00947-f002:**
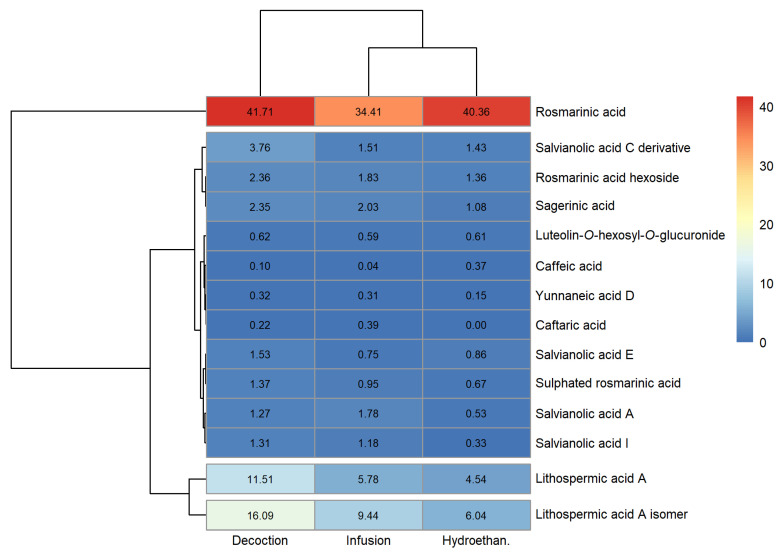
Clustered heatmap visualisation of phenolic compounds detected in lemon balm infusion, decoction and hydroethanolic extract (units: mg/g).

**Figure 3 foods-12-00947-f003:**
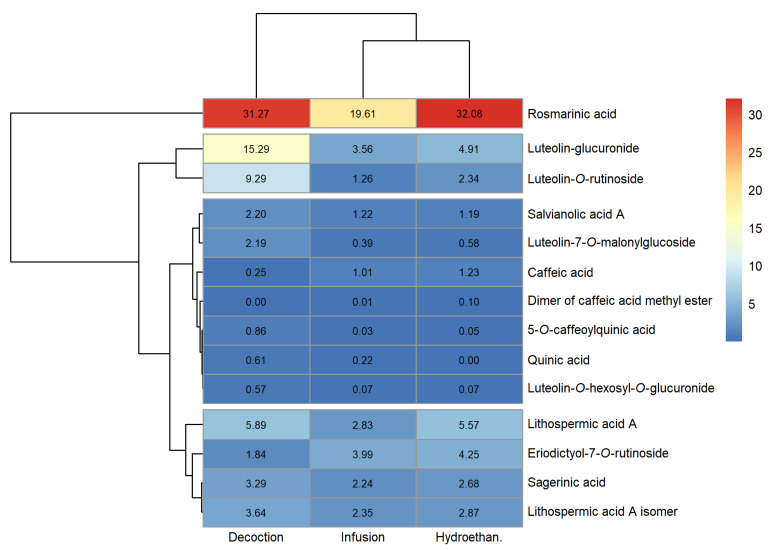
Clustered heatmap visualisation of phenolic compounds detected in spearmint infusion, decoction and hydroethanolic extract (units: mg/g).

**Table 1 foods-12-00947-t001:** Antibacterial activity of plant extracts expressed as minimum inhibitory concentration and minimum bactericidal concentration, MIC/MBC, respectively (mg/mL; mean ± SD, n = 3).

Extraction	Plant	SA ^1^	BC ^2^	LM ^3^	EC ^4^	ST ^5^	EntC ^6^
Infusion	Lemon balm	0.5/1	1/2	0.5/1	0.5/1	0.5/1	1/2
	Spearmint	0.5/1	0.5/1	0.5/1	0.5/1	0.5/1	1/2
	Sage	0.25/0.5	0.25/0.5	1/2	1/2	0.5/1	1/2
Decoction	Lemon balm	0.5/1	0.5/1	2/4	1/2	0.5/1	1/2
	Spearmint	0.5/1	0.5/1	1/2	0.5/1	0.5/1	1/2
	Sage	0.5/1	0.5/1	1/2	0.5/1	0.5/1	1/2
Hydroethanolic extraction	Lemon balm	0.5/1	0.5/1	1/2	0.5/1	0.5/1	0.5/1
	Spearmint	0.5/1	1/2	0.5/1	0.5/1	0.5/1	0.5/1
	Sage	0.5/1	1/2	0.5/1	0.5/1	0.5/1	0.5/1
E211 ^7^		4/4	0.5/0.5	1/2	1/2	1/2	2/4
E224 ^8^		1/1	2/4	0.5/1	0.5/1	1/1	0.5/0.5

Legend: ^1^ *S. aureus*, ^2^ *B. cereus*, ^3^ *L. monocytogenes*, ^4^ *E. coli*, ^5^ *Salmonella enterica* ser. Typhimurium, ^6^ *E. cloacae*, ^7^ Sodium benzoate, ^8^ Potassium metabisulfite.

**Table 2 foods-12-00947-t002:** Antifungal activity of plant extracts expressed as minimum inhibitory and minimum fungicidal concentration, MIC/MFC, respectively (mg/mL; mean ± SD, n = 3).

Extraction	Plant	AF ^1^	AN ^2^	AV ^3^	PF ^4^	PVC ^5^	TV ^6^
Infusion	Lemon balm	0.125/0.25	0.125/0.25	0.25/0.5	0.5/1	0.5/1	0.25/0.5
	Spearmint	0.125/0.25	0.25/0.5	0.25/0.5	0.5/1	1/2	0.25/0.5
	Sage	0.25/0.5	0.25/0.5	0.25/0.5	0.25/0.5	0.25/0.5	0.125/0.25
Decoction	Lemon balm	0.25/0.5	>4/>4	0.25/0.5	0.5/1	0.25/0.5	0.125/0.25
	Spearmint	0.25/0.5	>4/>4	0.25/0.5	0.5/1	0.25/0.5	0.25/0.5
	Sage	0.25/0.5	0.5/1	0.5/1	0.5/1	0.5/1	0.25/0.5
Hydroethanolic extraction	Lemon balm	0.5/1	0.5/1	0.25/0.5	0.25/0.5	0.25/0.5	0.25/0.5
	Spearmint	0.25/0.5	0.25/0.5	0.25/0.5	0.25/0.5	0.25/0.5	0.125/0.25
	Sage	0.5/1	0.5/1	0.25/0.5	0.25/0.5	0.125/0.25	0.125/0.25
E211 ^7^		1/2	1/2	2/2	1/2	2/4	1/2
E224 ^8^		1/1	1/1	1/1	0.5/0.5	1/1	0.5/0.5

Legend: ^1^ *A. fumigatus*, ^2^ *A. niger*, ^3^ *A. versicolor*, ^4^ *P. funiculosum*, ^5^ *P. verrucosum* var. *cyclopium*, ^6^ *T. viride*, ^7^ Sodium benzoate, ^8^ Potassium metabisulfite.

**Table 3 foods-12-00947-t003:** Antioxidant activity of plant extracts expressed as half-maximal inhibitory concentration (IC_50_, µg/mL) measured by the TBARS (mean ± SD, n = 9) and OxHLIA (mean ± SD, n = 3) essays.

Essay	Plant	Infusion	Decoction	Hydroethanolic Extract
TBARS ^1^	Lemon balm	125 ± 2.08 ^a^	204 ± 2.66 ^b^	206 ± 8.99 ^b^
	Spearmint	255 ± 11.0 ^c^	197 ± 5.68 ^a^	295 ± 9.77 ^c^
	Sage	235 ± 6.43 ^b^	196 ± 5.04 ^a^	132 ± 5.07 ^a^
OxHLIA ^2^ Δ*t* = 60 min	Lemon balm	61.4 ± 1.31 ^b^	27.0 ± 0.43 ^b^	13.5 ± 0.38 ^a^
	Spearmint	83.5 ± 1.84 ^c^	42.2 ± 0.62 ^c^	12.5 ± 0.17 ^a^
	Sage	21.9 ± 0.77 ^a^	8.93 ± 0.44 ^a^	23.9 ± 0.94 ^b^
OxHLIA ^2^ Δ*t* = 120 min	Lemon balm	95.5 ± 2.16 ^b^	41.6 ± 0.63 ^b^	27.4 ± 0.85 ^a^
	Spearmint	120 ± 1.84 ^c^	66.8 ± 0.92 ^c^	27.6 ± 1.28 ^a^
	Sage	38.4 ± 0.89 ^a^	23.5 ± 0.67 ^a^	56.4 ± 1.51 ^b^

Legend: ^1^ Thiobarbituric acid reactive substances, ^2^ Oxidative haemolysis inhibition essay. Trolox IC_50_ value: 5.4 ± 0.3 µg/mL (TBARS), 21.8 ± 0.25 µg/mL (OxHLIA Δ*t* = 60 min), 43.5 ± 1.00 µg/mL (OxHLIA Δ*t* = 120 min). For each essay, values with different superscript letters in a column mean significant differences (ANOVA, *p* < 0.05).

**Table 4 foods-12-00947-t004:** Anti-inflammatory activity of plant extracts expressed as half-maximal inhibitory concentration (IC_50_, µg/mL) measured by nitric oxide production inhibitory capacity (mean ± SD, n = 2).

Plant	Infusion	Decoction	Hydroethanolic Extract
Lemon balm	>400 ^b^	>400 ^b^	>400 ^b^
Spearmint	44.4 ± 0.66 ^a^	43.9 ± 4.26 ^a^	26.6 ± 1.65 ^a^
Sage	>400 ^b^	>400 ^b^	>400 ^b^

Dexamethasone IC_50_ value: 6 ± 1 µg/mL. Values with different superscript letters in a column mean significant differences (ANOVA, *p* < 0.05).

**Table 5 foods-12-00947-t005:** Cytotoxic activity of plant extracts expressed as half-maximal cell growth inhibitory concentration (GI_50_, µg/mL) measured by the sulforhodamine B essay (mean ± SD, n = 3).

Extraction	Plant	AGS ^1^	CaCo-2 ^2^	HeLa ^3^	MCF-7 ^4^	NCI-H460 ^5^	hFOB ^6^
Infusion	Lemon balm	215 ± 6.22 ^a^	290 ± 0.19 ^b^	249 ± 11.5 ^a^	239 ± 0.99 ^b^	>400	>400
	Spearmint	196 ± 7.44 ^a^	304 ± 0.55 ^c^	229 ± 21.2 ^a^	203 ± 1.50 ^a^	>400	>400
	Sage	249 ± 8.68 ^b^	242 ± 0.40 ^a^	248 ± 25.6 ^a^	198 ± 0.97 ^a^	>400	>400
Decoction	Lemon balm	255 ± 7.45 ^b^	>400 ^c^	301 ± 10.9 ^b^	>400	>400	>400
	Spearmint	258 ± 5.49 ^b^	396 ± 0.05 ^b^	289 ± 1.49 ^b^	>400	>400	>400
	Sage	215 ± 6.25 ^a^	269 ± 0.31 ^a^	111 ± 2.14 ^a^	320 ± 1.05 ^a^	>400	350 ± 4.25 ^a^
Hydroethanolic extract	Lemon balm	231 ± 2.75 ^b^	351 ± 3.30 ^c^	266 ± 11.5 ^b^	180 ± 4.43 ^a^	369 ± 3.37 ^a^	271 ± 2.52 ^a^
	Spearmint	162 ± 8.05 ^a^	285 ± 0.43 ^b^	215 ± 2.21 ^a^	210 ± 2.20 ^b^	381 ± 0.63 ^b^	264 ± 2.29 ^a^
	Sage	361 ± 3.74 ^c^	272 ± 0.06 ^a^	257 ± 1.17 ^b^	206 ± 2.34 ^b^	>400 ^c^	>400 ^b^

Legend: ^1^ Gastric adenocarcinoma, ^2^ Colorectal adenocarcinoma, ^3^ Cervical carcinoma, ^4^ Breast adenocarcinoma, ^5^ Large cell lung carcinoma, ^6^ non-tumour hFOB (human foetal osteoblasts). Ellipticine GI_50_ values: 1.23 ± 0.03 µg/mL (AGS), 1.21 ± 0.02 µg/mL (CaCo-2), 1.91 ± 0.12 µg/mL (HeLa), 1.02 ±0.02 µg/mL (MCF-7), 1.01 ± 0.01 µg/mL (NCI-H460) and 1.21 ± 0.08 µg/mL (hFOB). Values with different superscript letters in a column mean significant differences (ANOVA, *p* < 0.05).

## Data Availability

Summary data available upon request.

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
