# Peer review of "Phytochemical Composition and Bioactive Potential of *Melissa officinalis* L., *Salvia officinalis* L. and *Mentha spicata* L. Extracts"

_foods, 2023, doi:10.3390/foods12050947_

Round 1

Reviewer 1 Report

This manuscript characterized the polyphenolic profile and bioactive properties of infusions, decoctions and hydroethanolic extracts of three aromatic plants.

Please consider the following suggestions:

Please insert the aim of the study at the end of the Introduction section.

Please provide actual data from the papers that you compared your work to in the discussions section.

Reviewer 2 Report

1. The English need improvement since there are some grammatical and syntax errors in the manuscript. For example,

·         in line number 20, the word “replacement” may be as “the replacement”;

·         in line number 26, “ability” as “the ability”;

·         in line number 28, “Furthermore, and” as “Furthermore,”;

·         in line number 30, “on the” as “into the”;

·         in line number 31, “source” as “a source”;

·         in line number 92, “were added” as “was added”;

·         in line number 107, “Linear” as “a Linear”;

·         in line number 109, “carrier” as “a carrier”;

·         in line number 110, “capillary” as “a capillary”;

·         in line number 141, “decrease” as “a decrease”;

·         in line number 164, “delayed” as “the delayed”;

·         in line number 179, “allowed” as “was allowed”;

·         in line number 183, “Nitrite” as “The nitrite”;

·         in line number 186, “positive” as “a positive”;

·         in line number 211, “were produced” as “was produced”;

·         in line number 266, “which also” as “which were also”;

·         in line number 270, “highest” as “the highest”;

·         in line number 277, “greatest” as “the greatest”;

·         in line number 381, “allows to” as “allows us to”;

·         in line number 416, “greatest” as “the greatest”;

·         in line number 465, “insight on” as “insight into”;

·         in line number 471, “absence” as “an absence”.

The grammar mistakes which are not mentioned hecre are also to be checked and corrected properly.

2. There are some typing mistakes as well, and authors are advised to carefully proof-read the text. For example,

·         in line number 38, the word “increased” may be as “increasingly”;

·         in line number 181, “procedure it” as “procedure, it”;

·         in line number 238, “flavonoids” as “flavonoid”;

·         in line number 268, “belonging” as “belonged”;

·         in line number 281, “health promoting” as “health-promoting”;

·         in line number 435, “however some” as “however, some”.

The typos not mentioned here are also to be checked and corrected properly.

3. Check the abbreviations throughout the manuscript and introduce the abbreviation when the full word appears the first time in the abstract and the remaining for the text and then use only the abbreviation (For example, LPS, IC50, etc.,). Make a word abbreviated in the article that is repeated at least three times in the text, not all words  to be abbreviated.

4. Along with binominal classification of plants used in the present investigation should also be given with their respective family name.

5. The authors may include the details of the quantity of fresh sample used and the quantity of the powder obtained (the ratio between fresh plant and powder) under the heading “2.1. Plant Material and Extraction Procedures”.

6. The table legends should be improved and a proper footnote should be given. All legends should have enough description for a reader to understand the tables without having to refer back to the main text of the manuscript. For example, the necessary abbreviations should be given which are used in the present investigation.

7. The limitation of the present investigation may be given along with conclusion or under separate heading for understanding the concepts clearly.

Reviewer 3 Report

The manuscript "Phytochemical composition and bioactive potential of Melissa officinalis L., Salvia officinalis L., and Mentha spicata L. extracts" perform an exhaustive study of the bioactive properties of different plant extracts (including antibacterial/antifungal activity, antioxidant activity, anti-inflammatory activity, and cytotoxic activity), and identifies the phenolic compounds present in the extracts.

The study is well conducted, with an adequate methodology, and promising results for using plant extracts as substitutes for synthetic chemical additives in food, following the current trend of reducing the use of additives and looking for natural alternatives.

Comments to authors:

- Section 2.2: Indicate if the methodology used for the identification and quantification of phenolic compounds in the extracts is previously based on a validated methodology available. If so, provide the citation.

- A quantification of phenolic compounds in the extracts has been performed; however, it is not indicated in the text which standards have been used. It would be interesting to indicate in the supplementary material which standard has been employed for the quantification since not all of them are suitable for this purpose.

- The authors should provide the bibliographic references of the authors from whom the comparison of mass spectra has been made to carry out the identification.

- Line 111: 100-1500 m/z

- Section 2.3.1: The authors should clearly explain how they have done the antibacterial and antifungal activity assay: culture media used, incubation times... In addition, it is not specified how the determination of MBC has been carried out.

- Line 220-235: Be consistent with the number of decimal places provided in the quantification.

- Section 3.2.: In this section, I would include the tables that have been placed in the supplementary material (Antibacterial and antifungal activity), since the results obtained in the study are promising, and are directly correlated with the objective of the study.
